# The Impact of Patient-Reported Penicillin or Cephalosporin Allergy on the Occurrence of the Periprosthetic Joint Infection in Primary Knee and Hip Arthroplasty

**DOI:** 10.3390/antibiotics11101345

**Published:** 2022-10-02

**Authors:** Stella Stevoska, Verena Behm-Ferstl, Stephanie Zott, Christian Stadler, Sophie Schieder, Matthias Luger, Tobias Gotterbarm, Antonio Klasan

**Affiliations:** 1Department for Orthopaedics and Traumatology, Kepler University Hospital GmbH, 4020 Linz, Austria; 2Faculty of Medicine, Johannes Kepler University Linz, 4040 Linz, Austria; 3AUVA UKH Steiermark, 8020 Graz, Austria

**Keywords:** penicillin allergy, cephalosporin allergy, prophylactic antibiotic, knee arthroplasty, total hip arthroplasty, periprosthetic joint infections, revision

## Abstract

Reducing the risk of periprosthetic joint infections (PJI) requires a multi-pronged strategy including usage of a prophylactic antibiotic. A history of penicillin or cephalosporin allergy often leads to a change in prophylactic antibiotic regimen to avoid serious side effects. The purpose of the present retrospective study was to determine incidence of PJI based on perioperative antibiotic regimen in total hip arthroplasty (THA), total knee arthroplasty (TKA) or unicompartmental knee arthroplasty (UKA). A review of all primary THAs, primary TKAs and primary UKAs, undertaken between 2011 and 2020 in a tertiary referral hospital, was performed. The standard perioperative antibiotic for joint arthroplasty (JA) in the analyzed tertiary hospital is cefuroxime. There were no differences in prophylactic antibiotic regimen over time. In 7.9% (211 of 2666) of knee arthroplasties and in 6.0% (206 of 3419) of total hip arthroplasties, a second-line prophylactic antibiotic was used. There was no statistically significant higher occurrence of PJI between the first-line and second-line prophylactic antibiotic in knee arthroplasties (*p* = 0.403) as well as in total hip arthroplasties (*p* = 0.309). No relevant differences in age, American Society of Anesthesiologists (ASA) score and body mass index (BMI) between the groups were observed.

## 1. Introduction

Reducing the risk of periprosthetic joint infections (PJI) requires a multi-pronged strategy including usage of a prophylactic antibiotic [1,2,3]. The goal of the prophylactic antibiotic is to be present within the tissues at the time of initial incision and be maintained for the duration of the operation. It should reach adequate serum concentrations, above the minimum inhibitory concentrations for the organisms likely to be encountered during the operation [2,4]. A first- or second-generation cephalosporin (i.e., cefazolin or cefuroxime) offers an excellent distribution and excellent coverage across the spectrum of organisms responsible for most PJIs at a good cost-effectiveness and are therefore recommended for routine perioperative prophylaxis [2,4]. 

A history of penicillin or cephalosporin allergy, however, often leads to a change in prophylactic antibiotic regimen to avoid potential serious side effects due to cross-reactivity [2,4,5]. The use of vancomycin or clindamycin as second-line agents where cephalosporins are thought to be contraindicated is recommended. Clindamycin is the preferred alternative prophylactic antibiotic at institutions with low rates of methicillin-resistant *Staphylococcus aureus* (MRSA) infection [2], whereas vancomycin is selectively used in patients who have a high risk of MRSA colonization or are MRSA carriers [4,6].

Previous studies have demonstrated that reporting a penicillin or cephalosporin allergy and the consequent use of a second-line prophylactic antibiotic is associated with significantly higher rates of surgical site infections after various surgical procedures [5,7,8,9]. However, only a few studies have focused on the impact of patient-reported penicillin or cephalosporin allergy on the PJI rate in joint arthroplasty [5,8,10,11,12]. These studies were not unanimous on the effect of second-line prophylactic antibiotics on the occurrence of PJI [5,8,10,12,13]. The current evidence regarding the superiority of cephalosporin to non-cephalosporin alternatives therefore remains limited.

The aim of this retrospective clinical study was to identify and compare the occurrence of PJI in patients who received cefuroxime as a prophylactic antibiotic before primary arthroplasty with patients who, due to reported allergies, received a non-cefuroxime prophylactic antibiotic. Patients were stratified into two groups, knee arthroplasty and hip arthroplasty, and analyzed independently. Additionally, PJI onset time was identified and compared between the aforementioned cohorts according to usage of prophylactic antibiotics.

## 2. Results

A total of 3698 total hip arthroplasties (THAs) and 2876 unicondylar knee arthroplasties (UKAs) and total knee arthroplasties (TKAs) undertaken between 2011 and 2020 in a tertiary referral hospital were screened for inclusion in the study (Figure 1 and Figure 2). Cases with incomplete data or no follow-up were excluded, leaving 3419 THAs and 2666 UKAs and TKAs for further evaluation. Each group, THA and UKA/TKA, was divided into two additional cohorts—cefuroxime cohort and non-cefuroxime cohort—according to the use of a prophylactic antibiotic.

### 2.1. THA Group

In the present study, 7.7% (269 of 3419) of patients in the THA group reported a penicillin or cephalosporin allergy. Prophylactic antibiotic therapy with cefuroxime was given to 63 of 269 patients who reported the aforementioned allergies (23.4%). No allergic reactions were reported in those cases. In 206 of 3419 THAs (6.0%), due to reported allergies, a second-line (non-cefuroxime) prophylactic antibiotic was given (Figure 1, Table 1). Almost the entire non-cefuroxime cohort, 97.6% (201 of 206) of THAs, received clindamycin, only 1.9% (4 of 206) received ciprofloxacin and 0.5% (1 of 206) received vancomycin.

The demographics of both the cohorts, cefuroxime and non-cefuroxime, are shown in Table 1. The mean age of the patients in the THAs group was 65.9 years (SD 12.1) in the cefuroxime cohort and 67.8 years (SD 11.4) in the non-cefuroxime cohort (*p =* 0.028). Patients with reported allergies in the THAs group seemed to have a significantly higher mean BMI, 29.2 kg/m^2^ (SD 5.4), compared the mean BMI of the cefuroxime cohort, 28.0 kg/m^2^ (SD 4.8), with a *p* value of 0.001. (Table 1). Females reported more allergies to antibiotics, at 70.9% (*p* < 0.001) (Table 1). The mean American Association of Anesthesiology (ASA) score between cohorts was similar; however, ASA score III seemed to be significantly more frequent in the non-cefuroxime cohort at 40.8% compared to the cefuroxime cohort at 22.8% (*p* < 0.001). On the other hand, ASA scores I and II were significantly more common in the cefuroxime cohort, *p =* 0.002 and *p =* 0.003, respectively. There were no other relevant statistically significant differences between the cohorts observed.

### 2.2. UKA/TKA Group

In the UKA/TKA group, 10.1% (268 of 2666) of patients reported a penicillin or cephalosporin allergy. In 57 of 268 patients (21.3%), cefuroxime was given despite reported allergies. No allergic reactions were observed. A second-line, non-cefuroxime, prophylactic antibiotic was administered in 211 of 2666 (7.9%) UKAs and TKAs (Figure 2, Table 2). The entire non-cefuroxime cohort received clindamycin as a prophylactic antibiotic.

The demographics of both the cohorts, cefuroxime and non-cefuroxime, are shown in Table 2. The mean age of the patients in the UKAs and TKAs group was 68.7 years (SD 9.9) in the cefuroxime cohort and 69.1 years (SD 9.6) in the non-cefuroxime cohort (*p =* 0.572). As in the THA group, in the UKA/TKA group, females also reported a penicillin or cephalosporin allergy significantly more often, at 77.3% (*p* < 0.001) (Table 2). ASA score III was more often in the non-cefuroxime cohorts, at 34.6%, compared to cefuroxime cohorts, at 26.3% (*p =* 0.009) (Table 2). There were no other relevant statistically significant differences between the cohorts observed.

### 2.3. Revision and PJI Rate

Demographic characteristics and risk factors were similar in the two cohorts within each group. There was no statistically significant higher occurrence of revision between the first-line and second-line prophylactic antibiotic cohorts in knee arthroplasties (*p =* 0.757) and in total hip arthroplasties (*p =* 0.820). Furthermore, there was no statistically significant higher occurrence of revisions due to PJI between the cefuroxime and non-cefuroxime cohorts in primary knee arthroplasties (*p =* 0.403) and in primary hip arthroplasties (*p =* 0.309) (Table 3). Occurrence of PJI within the first 3 months after surgery was more common in hip arthroplasties, in 49 patients, than in knee arthroplasties, in 13 patients (Table 4).

## 3. Discussion

Penicillin or cephalosporin allergy was reported in 7.7% of the THA group and in 10.1% of the UKA/TKA group. The proportion of reported penicillin or cephalosporin allergies in the present study is slightly lower than in most previous studies, which reported proportions between 10% and 20% [5,8,13,14,15,16]. Females have generally reported more allergies to antibiotics, representing 70.9% and 77.3% of the non-cefuroxime cohorts in the THA group and UKA/TKA group, respectively (Table 1 and Table 2). This trend has already been observed in the cohorts of other studies [8,10,13].

Clindamycin was primarily the alternative antibiotic of choice in case of reported allergies. It was administered to 97.6% of patients in the non-cefuroxime cohort in the THA group and to the entire non-cefuroxime cohort in the UKA/TKA group. In the present study, in cases where cefuroxime was given despite reported antibiotic allergies, no allergic reactions were observed. This implies that the penicillin or cephalosporin allergies were over-diagnosed in the study population. As already shown in the previous studies, between 80% and 99% of patients with a reported allergy could be cleared to use cephalosporins after appropriate testing [5,9,17,18,19]. Furthermore, only 2.55% of patients with a confirmed penicillin allergy also have a confirmed cephalosporin allergy due to cross-reactivity [20,21]. Therefore, patient-reported allergies could be a notable contributor to the higher proportion of PJI due to the allergy-tailored choice of prophylactic antibiotics. There was no statistically significant higher occurrence of revision due to PJI between the first-line and second-line prophylactic antibiotic cohorts in knee arthroplasties (*p =* 0.403) or in total hip arthroplasties (*p =* 0.309) (Table 3). There are some studies to date that have analyzed the influence of different prophylactic antibiotic regimens on PJI rates after arthroplasty surgery [22,23,24,25,26]. However, not many have compared different prophylactic antibiotic regimens adjusted due to reported penicillin or cephalosporin allergy [5,8,10,12,13]. Wyles et al. [5] found higher rates of PJI when non-cephalosporin prophylactic antibiotics were used following primary TKA and THA. Buchalter et al. [12] also reported a significantly higher PJI rate after TKA in a non-cephalosporin cohort in their hospital. Nationwide research from Wu et al. [10] did not observe any significant differences in PJI rate between a cephalosporin and non-cephalosporin cohort after THA; they did however report a significant difference in PJI rate after TKA and after shoulder arthroplasty. In contrast to the present study, none of the aforementioned studies [5,10,12] described the timing or identity of prophylactic antibiotic in their non-cephalosporin cohorts. The exact proportion of the given second-line antibiotics, vancomycin and clindamycin, is thus unknown. Given that the mentioned studies [5,10,12] were conducted in the USA, where the prevalence of MRSA is higher, as in Europe [12,27], it is likely that the proportion of vancomycin administered was relevantly higher than in the present study. Correct dosing of vancomycin to achieve the required minimum inhibitory concentration is challenging, which may limit its effectiveness [28]. Incomplete administration [29] of vancomycin or vancomycin as monotherapy used in prophylaxis [28] is reported to have a greater risk for PJI. The use of vancomycin could therefore significantly contribute to the higher incidence of PJI in the non-cephalosporin cohorts of the aforementioned studies.

According to authors’ literature research, to date, only two studies have reported the incidence of PJI, where the majority of the allergic cohort received clindamycin as a second-line prophylactic antibiotic and are therefore comparable to the present study. However, the results of these studies are not uniform. On one hand, Stone et al. [8] reported no influence of second-line, non-cephalosporin, prophylactic antibiotic on the rate of SSI in total joint arthroplasty. In their cohort [8] of patients with reported penicillin or cephalosporin allergies, clindamycin was used in 80.7% and vancomycin in 12.4%. On the other hand, a report from the Swedish Knee Arthroplasty Register [13] found that the use of clindamycin as a prophylactic antibiotic instead of a beta-lactam antibiotic, such as cloxacillin, before primary TKA led to a 50% greater risk of infection. Clindamycin was believed to be less effective due to its bacteriostatic properties and high degree of intracellular distribution [13].

Additionally, the occurrence of PJI over time was observed in the present study. Most of PJIs were observed in the first three months (early PJI), more frequently in the HTEP group, at 73.1% (49 of 67 reported PJI cases), compared to 54.2% (13 of 24) in the UKA/TKA group. Robertsson et al. [13] reported in their study of Swedish Knee Arthroplasty Register an occurrence of PJI at 59% within the first three months and 79% occurring within the first year. Wyles et al. [5] demonstrated a significantly higher infection-free survivorship among arthroplasties, THA and TKA, receiving cefazolin compared to the non-cefazolin cohort. The high incidence of PJI after THA within the first three months in this study is worrisome; however, there were no significant differences in PJI rate within first three months between the cefuroxime and non-cefuroxime cohorts in the UKA/TKA group or in the THA group (Table 4). Therefore, the higher occurrence of early PJI after THA is most likely not the fault of the antibiotic prophylaxis. Other procedures in preoperative, intraoperative or early postoperative management should be re-evaluated. No other studies that investigated the influence of reported allergies have reported about the occurrence of PJI over time. Follow-up in these studies was mostly limited to three months.

The present study is the only study that used cefuroxime, a second-generation cephalosporin, as a first-line prophylactic antibiotic. Most of the aforementioned studies [5,8,10,12] used cefazolin, a first-generation cephalosporin. PJI rates in their cefazolin cohorts were between 0.5% and 0.6% in first 3 months [8,12] and around 1% in first year [10]. In the present study, there was a higher occurrence of early PJI (first 3 months) after THA in the cefuroxime cohort, 1.4%, compared to cefazolin cohorts of previous studies. Nevertheless, the PJI rates after TKA in the present cefuroxime cohort (0.5% in the first 3 months and 0.7% in the first year) were comparable to the PJI rates in the previous cefazolin cohorts. Based on the similar coverage spectra of cefazolin and cefuroxime, as well as similar PJI rates between the studies, it is unlikely that different choice of first-line prophylactic antibiotic could significantly influence the results of the present study. There are many studies comparing different first-line prophylactic antibiotics—cefuroxime, cefazolin and cloxacillin—in vascular and neurosurgery [30,31]; however, the current literature on this topic in the orthopedic field remains scarce.

This study must be interpreted in light of important limitations. Owing to its retrospective design, there is a collection and selection bias. Due to follow-up or incomplete data, 489 patients were lost. It is possible that complications in some patients with small follow-up were treated in other hospitals. However, the analyzed institution is the biggest tertiary referral hospital in the region; therefore, not many cases are expected to be treated elsewhere. Secondly, there might be a change in the hygiene protocol or in the operating theaters used in the included years, unknown to authors of this study. However, a single-institution design limits some possible confounding variables. On the other hand, the limited number of patients due to the single-institution design and low overall infection rate could influence the detection of the prophylaxis effect on PJI occurrence. A statistical power analysis, however, did not confirm that suspicion. Moreover, in register studies where findings depend on the accuracy of coding, an over-diagnosis of PJI is possible. Thirdly, a lack of bacterial growth, in the present study defined as an aseptic revision, does not necessarily imply surgical site sterility. However, no significant differences in the rate of revision due to causes other than infection, depending on which prophylactic antibiotic was used, were found in the present study.

## 4. Methods 

This retrospective, single-center cohort study was approved by the University’s Ethics Board (1156/2022). All elective primary TKAs, primary UKAs and primary THAs, undertaken between 2011 and 2020 in a tertiary referral hospital, were identified. Cases with insufficient data were excluded from the study (Figure 1 and Figure 2). Retrospective information was collected on demographics (age and gender), body mass index (BMI), drug allergies, preoperative performance status (measured by use of the American Association of Anesthesiology (ASA) classification), laterality, type of implant (UKA, TKA or THA), choice of prophylactic antibiotic treatment and potential revision surgery. In case of revision, the microbiological database was searched for the pathogens detected intraoperatively. Cases in which the same organism has been identified in at least two cultures or the sinus tract has been present were considered to meet the consensus criteria for PJI [32]. Patients meeting other criteria established by the Musculoskeletal Infections Society (MSIS) [32] were excluded. Due to different rates of PJI depending on the joint to be replaced [33], a stratification into two groups, THA and UKA/TKA, and an independent analysis was performed. 

### 4.1. Surgical Treatment 

There were no variations in the preoperative washing protocol, the method of skin preparation, the hand hygiene solutions used or the type of sterilization of surgical equipment and no major differences in surgical approach throughout the study period. MRSA colonization status was not assessed. All patients presenting with possible PJI due to an elevated CRP and/or signs of local inflammation underwent a sterile aspiration of synovial fluid and microbiological analysis. Depending on the time of infection and clinical findings’ debridement and implant retention (DAIR), one- or two-stage revision arthroplasty, followed by long-term antibiotic suppression, was performed. The revision techniques for PJI remained the same throughout the study period.

### 4.2. Prophylactic Antibiotic Treatment 

The chosen prophylactic antibiotic used before the primary, non-revision surgery throughout the study period was a single shot of 1500 mg intravenous cefuroxime. In the case of history of penicillin and/or cephalosporin hypersensitivity, antibiotic prophylaxis was adapted accordingly and was usually replaced by a single preoperative shot of clindamycin, either 300 mg or 600 mg. Antibiotics were administered intravenously during the induction of anesthesia, 30 min before the surgical incision.

### 4.3. Microbiology 

In case of a possible PJI, routinely, a sterile aspiration of synovial fluid was performed preoperatively, and at least three tissue samples and one sample of synovial fluid were obtained for culture intraoperatively. Synovial fluid was aseptically inoculated into aerobic and anaerobic blood culture bottles (BACTEC, Beckton Dickinson, Franklin Lakes, NJ, USA) and incubated for 14 days. Tissue samples were obtained from the subfascial tissue and the proximal and distal interfaces of the prosthesis. Each culture was transported, processed and analyzed according to international standards and the definitions of the European Committee on Antimicrobial Susceptibility Testing (EUCAST) [34]. Tissues were inoculated onto Columbia agar, chocolate agar, McConkey agar and Schädler agar and into brain heart infusion (BHI) broth. The broth was subcultured the next day. Additionally, sonication of the explanted prosthetic components was performed. The supernatant was aspirated, and the sediment was inoculated into the BHI broth. All culture media were incubated for 14 days and inspected every day for bacterial growth. There were no major differences in tissue sampling between UKAs, TKAs and THAs over time.

### 4.4. Outcome Measures

The main outcome measure of this retrospective study was to identify and compare the occurrence of PJI in patients who received cefuroxime as a prophylactic antibiotic before primary arthroplasty with patients who, due to reported allergies, received a non-cefuroxime prophylactic antibiotic. Additionally, PJI onset time was identified and compared between the aforementioned groups.

### 4.5. Statistical Analysis

Data were collected using Microsoft Excel (Microsoft Corporation, Redmond, WA, USA) and analyzed with SPSS v.24 (IBM Corp., Armonk, NY, USA). The study cohort was described using descriptive statistics. Categorical variables were expressed as percentages of the total sample for those variable and continuous variables as means and standard deviation and follow-up time as median and interquartile range. A post hoc power analysis was performed, assuming non-inferiority between the groups. Using the revision rates and differences observed in the study, with an alpha of 0.05 and power of 0.9 and with the non-inferiority limit set to 2%, a sample size of 1610 was needed. Overall revision rate and PJI rate were calculated by determining the number of cases where revision and PJI occur. The differences between the groups were analyzed using the chi-squared test. The presence of an interaction and the role of confounding factors were evaluated. A *p*-value of <0.05 was considered statistically significant.

## 5. Conclusions

There are no implications that the usage of clindamycin, as the second-line antibiotic, increases risk for PJI. However, the choice of prophylactic antibiotic could affect the pathogen spectrum and antibiotic resistance. Further analysis should be performed. 

## Figures and Tables

**Figure 1 antibiotics-11-01345-f001:**
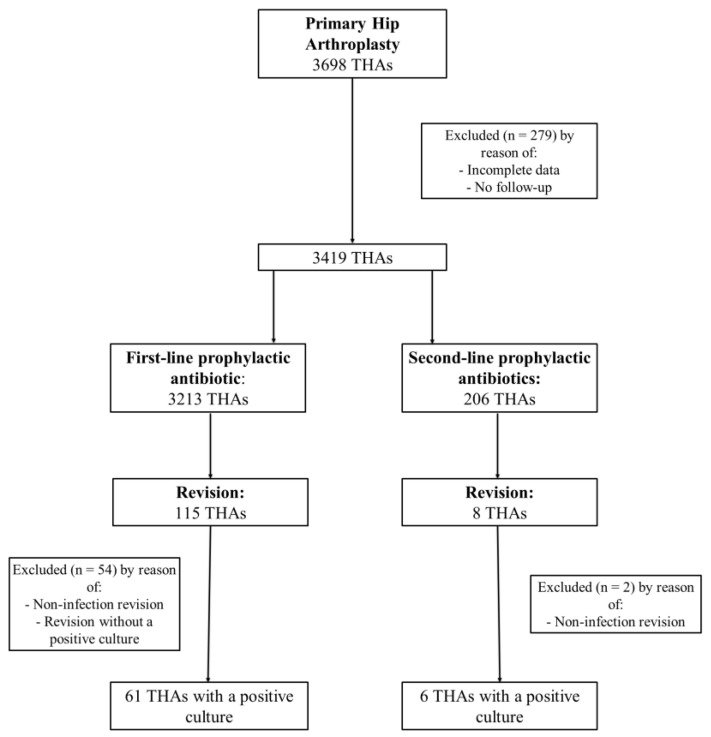
Flowchart of patients’ inclusion criteria in THA group.

**Figure 2 antibiotics-11-01345-f002:**
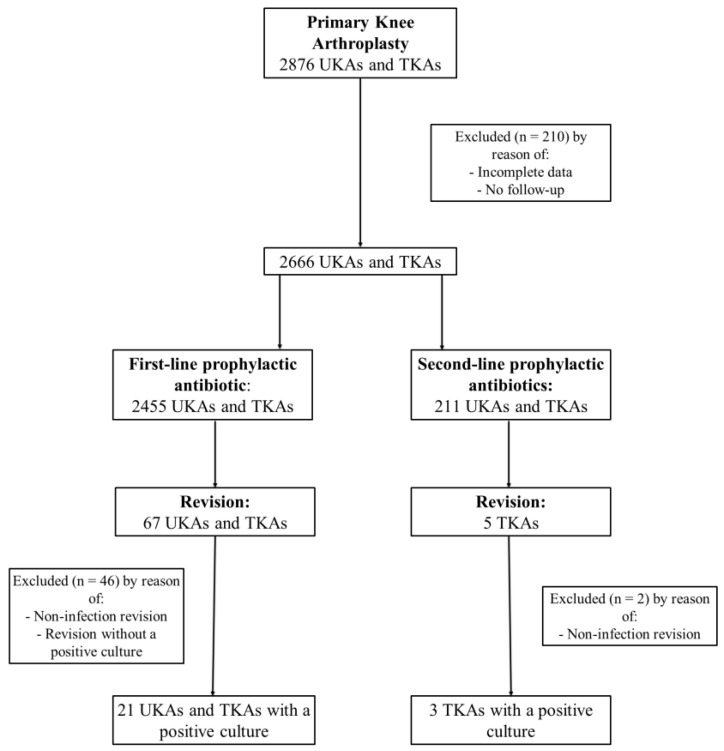
Flowchart of patients’ inclusion criteria in UKA and TKA group.

**Table 1 antibiotics-11-01345-t001:** Demographics in total hip arthroplasty group according to prophylactic antibiotic.

Variable	Cefuroxime (*n* = 3213)	Non-Cefuroxime (*n* = 206)	*p*-Value (Significant < 0.05)
Mean age, years (SD)	65.9 (12.1)	67.8 (11.4)	0.028
Gender, *n* (%):			
Female	1707 (53.1)	146 (70.9)	<0.001
Male	1506 (46.9)	60 (29.1)	
Laterality, *n* (%):			
Left	1503 (46.8)	103 (50.0)	0.370
Right	1681 (52.3)	103 (50.0)	
Both	29 (0.9)	0	
Mean BMI, kg/m^2^ (SD)	28.0 (4.8)	29.2 (5.4)	0.001
Mean ASA score	2.1	2.4	
ASA score, *n* (%):			
I	508 (15.8)	16 (7.8)	0.002
II	1953 (60.8)	104 (50.5)	0.003
III	733 (22.8)	84 (40.8)	<0.001
IV	19 (0.6)	2 (1.0)	0.500
Median Follow-up, months (IQR)	15.9 (39.1)	15.4 (35.1)	0.833

BMI, body mass index; ASA, American Society of Anesthesiologists; SD, standard deviation.

**Table 2 antibiotics-11-01345-t002:** Demographics in knee arthroplasty group according to prophylactic antibiotic.

Variable	Cefuroxime (*n* = 2455)	Non-Cefuroxime (*n* = 211)	*p*-Value (Significant < 0.05)
Mean age, years (SD)	68.7 (9.9)	69.1 (9.6)	0.572
Gender, *n* (%):			
Female	1492 (60.8)	163 (77.3)	<0.001
Male	963 (39.2)	48 (22.7)	
Laterality, *n* (%):			
Left	1188 (48.4)	105 (49.8)	0.700
Right	1243 (50.6)	106 (50.2)	
Both	24 (1.0)	0	
Arthroplasty, *n* (%)			
TKA	1668 (67.9)	164 (77.7)	0.003
UKA	787 (32.1)	47 (22.3)	
Mean BMI, kg/m^2^ (SD)	29.9 (5.4)	30.3 (6.1)	0.307
Mean ASA score	2.2	2.8	
ASA score, *n* (%):			
I	212 (8.6)	11 (5.2)	0.085
II	1591 (64.8)	125 (59.2)	0.110
III	645 (26.3)	73 (34.6)	0.009
IV	7 (0.3)	2 (1.0)	0.110
Median Follow-up, months (IQR)	13.1 (23.6)	13.5 (23.8)	0.757

TKA, total knee arthroplasty; UKA, unicompartmental knee arthroplasty; BMI, body mass index; ASA, American Society of Anesthesiologists; SD, standard deviation.

**Table 3 antibiotics-11-01345-t003:** Revision rate and PJI rate in knee arthroplasty group and total hip arthroplasty group according to prophylactic antibiotic.

Group	Variable	Cefuroxime	Non-Cefuroxime	*p*-Value
UKA/TKA	Revision rate, *n* (%)	67/2455 (2.7)	5/211 (2.4)	0.757
	PJI rate, *n* (%)	21/2455 (0.9)	3/211 (1.4)	0.403
THA	Revision rate, *n* (%)	115/3213 (3.6)	8/206 (3.9)	0.820
	PJI rate, *n* (%)	61/3123 (2.0)	6/206 (2.9)	0.309

TKA, total knee arthroplasty; UKA, unicompartmental knee arthroplasty; THA, total hip arthroplasty; *n*, number; PJI, periprosthetic joint infection.

**Table 4 antibiotics-11-01345-t004:** Occurrence of PJI (%) over time in knee arthroplasty group and total hip arthroplasty group according to prophylactic antibiotic.

Group	Prophylactic Antibiotic	1 Month	3 Months	<1 Year	>1 Year
UKA/TKA	Cefuroxime (*n* = 21), *n* (%)	7 (33.3)	5 (23.8)	5 (23.8)	4 (19.1)
	Non-cefuroxime (*n* = 3), *n* (%)	1 (33.3)	0 (0.0)	0 (0.0)	2 (66.7)
THA	Cefuroxime (*n* = 61), *n* (%)	33 (54.1)	12 (36.4)	2 (3.3)	14 (23.0)
	Non-cefuroxime (*n* = 6), *n* (%)	3 (50.0)	1 (16.7)	2 (33.3)	0 (0.0.)

TKA, total knee arthroplasty; UKA, unicompartmental knee arthroplasty; THA, total hip arthroplasty; *n*, number.

## Data Availability

The data presented in this study are available upon request from the corresponding author.

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
