# Peer review of "The Impact of Patient-Reported Penicillin or Cephalosporin Allergy on the Occurrence of the Periprosthetic Joint Infection in Primary Knee and Hip Arthroplasty"

_antibiotics, 2022, doi:10.3390/antibiotics11101345_

Round 1

Reviewer 1 Report

This is a retrospective clinical study, the principal aim was to identify and compare the occurrence of PJI in patients who received cefuroxime as a prophylactic antibiotic before primary arthroplasty with patients who due to reported allergies received a non-cefuroxime (generally clyndamicin) as prophylactic antibiotic.

The results with figures and tables are clearly presented.

The main considerations that must be made before considering the article for publication are:

The definition of PJI that they use: Cases in which an organism has been identified were considered to meet the consensus criteria for PJI, does not conform to the established standards. The Parvisi study to which they refer or the IDSA guidelines establish that for a higher criterion, 2 positive cultures for the same microorganism are necessary. We believe that the PJI criteria used should be better specified, showing, if possible, how many patients had major criteria. The number of patients included in the non-cephalosporin antibiotic prophylaxis arm is probably insufficient to demonstrate differences. Due to the small number of patients, they cannot adjust for the differences observed between the study arms: gender, ASA.

As less important consideration: 

I think the abbreviations used in the article should be specified.

It would be important to show the results of the microbiological isolations in a table to see if there are differences in the isolated microorganisms according to the prophylactic treatment administered.

They use as prophylaxis, in patients with allergy clindamycin, an antibiotic rarely used in surgical prophylaxis. It is important not to extrapolate it to other antibiotics, this should be stated in the discussion.

Reviewer 2 Report

      Good Introduction, results , discussion and conclusion of impact of patient reported cephalosporin or penicillin allergy on occurrence of periprosthetic joint infection. Good (non-inferiority) conclusions drawn in an area not researched before.

The research’s main question is the impact of patient-reported penicillin or cephalosporin al- 2 lergy on the occurrence of the periprosthetic joint infection in 3 primary knee and hip arthroplasty.

The manuscript is an important area of antibiotic prophylaxis that has not been extensively studied.

I think conclusions drawn are appropriate and references are appropriate and extensive.   I would place Methods section after introduction and before results and discussion. Besides, I have no other additional comments.
